# Lymphaticovenous Anastomosis for Age-Related Lymphedema

**DOI:** 10.3390/jcm10215129

**Published:** 2021-10-31

**Authors:** Shuhei Yoshida, Isao Koshima, Hirofumi Imai, Solji Roh, Toshiro Mese, Toshio Uchiki, Ayano Sasaki, Shogo Nagamatsu

**Affiliations:** 1The International Center for Lymphedema, Hiroshima University Hospital, 1-2-3, Kasumi, Minami-ku, Hiroshima 734-8551, Japan; koushimaipla@gmail.com (I.K.); imaih61@hiroshima-u.ac.jp (H.I.); solji6004@yahoo.co.jp (S.R.); mese.toshiro1818@me.com (T.M.); 2Department of Plastic and Reconstructive Surgery, Hiroshima University, 1-2-3, Kasumi, Minami-ku, Hiroshima 739-8551, Japan; toshio.uchiki@gmail.com (T.U.); vin.pichon.ayano@gmail.com (A.S.); shogonagamatsu@gmail.com (S.N.)

**Keywords:** lymphedema, aging, lymphaticovenous anastomosis

## Abstract

Introduction: Primary lymphedema is usually caused by intrinsic disruption or genetic damage to the lymphatics but may also be the result of age-related deterioration of the lymphatics. The aims of this study were to determine the characteristics of age-related lymphedema and to assess the effectiveness of lymphaticovenous anastomosis (LVA) in its treatment. Methods: Eighty-six patients with primary lymphedema affecting 150 lower limbs were divided into three groups according to whether the age of onset was younger than 35 years, 35–64 years, or 65 years or older. Indocyanine green (ICG) lymphography was performed, followed by LVA surgery. ICG lymphography images were visually classified according to whether the pattern was linear, low enhancement (LE), distal dermal backflow (dDB), or extended dermal backflow (eDB). The lower extremity lymphedema (LEL) index score was calculated before and after LVA. Lymphatic vessel diameter and detection rates were also recorded. Results: In the ≥65 group, the lymphedema was bilateral in 54 patients and unilateral in 1 patient. There was statistically significant deterioration in the LEL index score with progression from the linear, LE, dDB through to the eDB pattern in the ≥65 group. The lymphatic vessel diameter was significantly greater in the ≥65 group. The rate of improvement was highest in the ≥65 group. Conclusion: Age-related lymphedema was bilateral and deterioration started distally. The lymphatic vessels in patients with age-related lymphedema tended to be ectatic, which is advantageous for LVA and may increase the improvement rate.

## 1. Introduction

Primary lymphedema with onset after 35 years of age is known as lymphedema tarda and is generally caused by intrinsic disruption or genetic damage in the lymphatic drainage system. Some evidence suggests that lymphedema could also be caused by an age-related decline in lymphatic pump function [1,2,3,4,5]. However, the characteristics of age-related lymphedema are not well understood and no treatment guidelines for this condition have been established.

The aims of this study were threefold: to determine whether aging could be a cause of lymphedema in patients with onset after age 65 years, which is defined as “elderly” by the World Health Organization [6]; to clarify its characteristics of these patients; and to assess the effectiveness of the now widely used surgical method of lymphaticovenous anastomosis (LVA) [7,8,9] in these patients.

## 2. Patients and Methods

This retrospective study included patients with lower limb lymphedema treated by LVA at Hiroshima University Hospital from May 2017 to December 2018. The study was approved by our institutional review board (approval number: E-1413) and conducted reference to the Declaration of Helsinki and the STROBE guidelines (http://www.strobe-statement.org, accessed on 7 April 2017). All patients submitted written informed consent.

The exclusion criteria were need for support/auxiliary aids for ambulation; body mass index (BMI) ≥ 35; edema controllable by conventional compression therapy using elastic stockings; history of a major invasive procedure to treat pelvic cancer including uterine cancer, prostate cancer, rectal cancer, metastatic cancer, and inguinal lymphadenectomy, radiation therapy to the lower limbs or abdomen, chronic cardiac failure, chronic kidney failure, hepatic cirrhosis, hypoproteinemia, deep venous thrombosis, phlebostasis or venous reflux, thyroid disorder, edema caused by other endocrine disorder, drug-related edema, or arteriovenous malformation. Preoperatively, the compression stocking pressure was adjusted to between 18 and 32 mmHg at the highest tolerable pressure.

ICG lymphography were performed in all patients. First, to assess lymphatic function by measuring transit time, which is the time required for the dye to reach the lymph nodes in the groin [10]. During this procedure, the patient lay in the supine position and repeated plantar flexion and dorsiflexion movements of the ankles and toes to reduce the delay in flow that often occurs during passage across the ankle joints and affects the ICG lymphography images acquired [11]. Lymphoscintigraphy was also performed when transit time was ≥10 min despite lying supine and performing these regular movements [12]. A diagnosis of lymphedema was made based on an abnormal lymphoscintigram with delayed transit of radiolabeled colloid (i.e., longer than 50 min) [13,14,15,16,17] considering the existence of lower limb edema although there are no causes of edema except for delayed lymphatic transport.

The patients were divided into three groups according to their age of onset, younger than 35 years, 35–64 years, or 65 years or older. ICG lymphography images were recorded in the plateau phase (i.e., 12–18 h after the injection or on the following day). The ICG image patterns were then classified according to their laterality, whether to bilateral or unilateral, and were then classified according to whether the pattern seen as linear (the fluorescent lymphatic vessels were seen as a linear pattern between the injection site and the inguinal nodes), low enhancement (LE; a linear pattern is observed only around the ankle joint without enhancement in the proximal portion), distal dermal backflow (dDB; deterioration in the enhanced lymphatics only in the distal portion of the lower extremity below the knee), or extended dermal backflow (eDB; deterioration extended over the knee to the groin) as shown in Appendix A. Obtained ICG images were reviewed by two plastic surgeons working independently [18].

LVAs were performed under local anesthesia in all cases along linear pattern or along the greater saphenous vein course in the area of DB pattern or no enhancement. The LVA procedures were an end-to-end manner using 11-0 or 12-0 nylon micro sutures under a surgical microscopes. LVAs are performed by two surgeons simultaneously in the same way previously reported [18] within the limited time of 3 h. The number of LVAs was determined by this time limit. The success of the LVAs was reviewed by two plastic surgeons working independently to reduce the observer’s bias.

The laterality of the lymphedema, the time interval between age of onset and LVA surgery, and the number of LVAs were recorded. A microscale (Kono Seisakusyo Co., Ltd., Ichikawa, Japan) was used to measure the diameter of each lymphatic vessels. Measurements were performed by the two plastic surgeons working independently. The mean vessel diameter for each extremity was calculated. The lymphatic detection rate in each extremity was calculated by dividing the total number of skin incision cites for each extremity by the number of cite where at least one lymphatic suitable for anastomosis was detected. Compression therapy was resumed in the same way as before surgery 1 week after LVA.

The lower extremity lymphedema (LEL) index was calculated [19]. Circumferences were measured at the foot, ankle, knee, 10 cm below the knee, and 10 cm above the knee for each extremity with the patient in the supine position. Sum of the squares of the circumference in five areas was divided by the BMI. A lower score of LEL index indicates less severe edema. All measurements were performed in the outpatient clinic in the morning by a nurse not involved in the research and randomly assigned to each patient on the day of measurement.

The postoperative reduction in volume was calculated as follows:

[{(preoperative LEL) − (postoperative LEL)}/(preoperative LEL)] × 100 (%) = the improvement rate.

### Statistical Analysis

Data are shown as the mean and standard deviation (range). The LEL index and lymphatic vessel diameters recorded in the ≥65 group were compared according to type of lymphography pattern using the Tukey–Kramer test. The laterality of the lymphedema and the distribution of the ICG pattern were compared between the <35 group, 35–64 group, and ≥65 group using the chi-squared test.

The LEL index, lymphatic detection rate, lymphatic vessel diameter, number of LVAs performed, number of skin incisions for LVA, rate of improvement in the LEL index, and interval between age of onset and LVA surgery were compared between the three age groups using the Tukey–Kramer test. BMI was compared according to the type of lymphography pattern independent of age using the Tukey–Kramer test. All statistical analyses were performed using Statcel 4 software (OMS Publishing, Inc., Tokyo, Japan). A *p*-value of less than 0.05 was considered statistically significant.

## 3. Results

This study included an initial 490 patients with lower extremity lymphedema who underwent lymphedema surgery at our institution between May 2017 and December 2018. Among them, 4 patients in the <35 group, 189 patients in the 35–64 group, and 211 patients in the ≥65 group were excluded by the exclusion criteria (Appendix A). Eighty-six patients with 150 edematous lower limbs who showed delayed transit time on both ICG lymphography and lymphoscintigraphy were enrolled in the study. There were no complications such as lymphorrhea or delayed wound healing during LVA procedures and postoperative course. None of the patients developed cellulitis during the study period. None of the patients had a negative ICG study. All patients had a diagnosis of lymphedema or edema caused by lymphatic stasis or delay. According to the International Society of Lymphology scoring system for lymphedema [20], 42.7% had stage I, 44% had stage II, and 13.3% had stage III (Table 1). Most of the patients in the <35 group were older than 10 years of age with the exception of 2 cases (Appendix A).

There was significant deterioration in the LEL index (Figure 1) and a significant decrease in lymphatic vessel diameter (Figure 2) with progression from the linear, LE, dDB through to the eDB pattern in the ≥65 group. A bilateral pattern was found significantly more frequently in older patients than in younger patients (Figure 3). We detected the eDB pattern in most patients in the <35 group; however, other types of ICG pattern were seen in the older patients (Figure 4). There was statistically significant deterioration in the LEL index with decreasing age of onset (Figure 5).

The lymphatic detection rate (Figure 6), lymphatic vessel diameter (Figure 7), and number of LVAs performed increased significantly with increasing age of onset (Appendix A), although there was no significant difference in the number of skin incisions for LVA among the groups (Appendix A). The rate of improvement in the LEL index after LVA was significantly greater in older patients (Figure 8). The interval between the age of onset and LVA surgery decreased significantly with increasing age (Figure 9). Independent of age, BMI was highest in the LE pattern (Appendix A).

## 4. Discussion

Leg edema is a common problem in older patients and has a wide range of possible causes. The differential diagnosis includes systemic illness such as chronic cardiac failure, hepatic cirrhosis, hypoproteinemia, or a thyroid disorder, a pelvic invasion, or venous insufficiency, and various drug-related edema [21]. In this study, we carefully excluded these causes and diagnosed lymphedema based on delayed lymphatic transport seen on ICG lymphography. In general, lymphedema with no obvious cause is classified into primary lymphedema, even when the onset is in adulthood. However, there are various features and cause in primary lymphedema. In our study, the features of primary lymphedema varied according to age of onset. First, the patients younger than 35 years in our study had a tendency to be unilateral while bilateral in older patients. Second, the lymphatic diameter was greater in older patients in the early stage than in younger patients or older patients in the advanced stage. Therefore, it was reasonable to believe that primary lymphedema in our study has varied causes depend on its onset age. Besides, it is also inferred that lymphatic pump function was deteriorated by age-related causes and the bilateral primary lymphedema appeared in the older patients in this study.

Decreases in contraction frequency [3], pumping activity, systolic lymph flow velocity [4], and lymphatic vessel density [22] have been demonstrated in animal models of aging lymphatic collecting ducts. Aging is speculated to be dominant a chronic inflammatory process [23,24,25]. This progressed chronic inflammatory process would cause deleterious vascular changes including lymphatic pump function in aging [3]. Furthermore, lymph drainage has been shown to decrease with advancing age in clinical studies using lymphoscintigraphy [1] and ICG lymphography [5]. One animal study found that older animals have significantly greater lymphatic vessel diameters than younger adults at the same anatomical location [26]. It can be speculated that the same pathophysiological changes occur in humans. In Japanese population, mean diameter of lymphatics in the lower extremities is around 0.5 mm in our empirical data [27]. Therefore, we speculate that a functional disorder of the lymphatics may be a common cause of edematous lower extremities in older patients.

In this study, the characteristics of age-related lymphedema were bilateral involvement and deterioration in the LEL index with progression from the linear, LE, dDB through to the eDB pattern on ICG lymphography. These features indicate that lymphedema progressed from the distal region and not from the proximal area, which is different from the course of deterioration that occurs with secondary lymphedema caused by an obstruction such as pelvic cancer [28]. In secondary lymphedema, lymphatic function is initially retained; however, lymph then starts to accumulate in the proximal region [29] and extends distally. In contrast, age-related lymphatic deterioration is suspected to develop across the entire area. The contractile force in the lymphatics also decreases, leading to a reduction in the amount of lymph that can be transported in the lymphatics. Lymph then starts to accumulate in the distal region owing to the gravity effect, unlike in secondary lymphedema associated with an obstruction.

As reported previously [30,31], our younger cases of lymphedema tended to be unilateral rather than bilateral. Schook et al. found that lymphedema was more likely to be bilateral in patients presenting in infancy or childhood than in those presenting in adolescence and that unilateral involvement was more likely in patients whose onset was during adolescence [31]. Their findings correspond well with the results of our study.

Furthermore, the younger patients were significantly worse than older patients in LEL index with a DB pattern in both the lower leg and thigh areas. It is difficult to determine whether these features reflect a congenital or acquired physiological defect in younger patients with unilateral lymphedema. However, it is assumed that these patients have congenitally vulnerable lymphatic vessels and that minor events, for example, an inflammatory reaction or mild trauma cause lymphedema that is too subtle to be classified as secondary lymphedema. This type of impaired lymphatic function is more likely to be secondary to errors in fetal development, although the lymphedema in the younger patients in our study may have been an acquired insufficiency. Alternatively, an insufficiency may exist from birth and compensated for by the lymphatic system until there is subtle invasion resulting in overload of lymphatic transport capacity [32,33,34]. Therefore, our younger patients with unilateral lymphedema were likely to have a combination of secondary and congenital characteristics that resulted in more severe lymphedema.

LVA is assumed to be easier to perform in older patients because their lymphatic vessels tend to be ectatic, which results in a high detection rate and enables more LVAs to be performed. In a previous study, patients with secondary lymphedema were found to have more ectatic lymphatic vessels because endolymphatic pressure increased with worsening lymphedema [35]. As the secondary lymphedema progressed, smooth muscle cells in the lymphatics were transformed into collagen fibers, leading to irreversible contraction of these vessels. In the terminal stage of lymphedema, lymphatics lose their transport capacity and become either narrowed or completely obstructed [35]. However, in age-related lymphedema, the decreases in contraction frequency and activity cause pathophysiologic changes, so that the lymphatic vessels are dilated from an early stage. As the lymphedema progresses, smooth muscle cells in the lymphatics are transformed into collagen fibers, which leads to irreversible contraction of the lymphatics in the same way as in secondary lymphedema. In our study, the lymphatics in these patients were initially dilated when the linear or LE pattern was present; however, as the lymphedema progressed (with worsening of the LEL index value), the lymphatic diameter decreased in the order of dDB to eDB until the end stage of lymphedema, as in the unilateral pattern seen in younger patients. LVA is best performed in the early stages, even in patients with age-related lymphedema [36,37,38]. We believe that age-related lymphedema should be treated because of its progressive nature and have focused more recently on prevention and risk reduction [39].

In the present study, patients <35 years of age had the most severe disease features and the least favorable surgical outcomes. Furthermore, our younger patients had longer interval between disease onset and LVA. This pronounced tendency has yet to be explained but may lead to smaller diameter of lymphatics and worse outcomes owing to long lasting chronic damage to lymphatics.

The LE pattern is often seen on ICG lymphography in patients with a high BMI [40]. In this study, patients with this pattern also had the highest BMI; however, a significant difference in the ICG pattern was seen only between LE and eDB and between dDB and eDB, which may be explained by the fact that patients with a BMI ≥35 were excluded from the study. According to previous reports, the diameter of lymphatic was significantly greater in the obese group than in the nonobese group [40]. The results are consistent with studies in hypercholesterolemic mice [41,42]. The mechanism how obesity impairs the lymphatic function was unknown. However, a hypothesis is that the amount of lymph produced by obesity patients increases as BMI increases and the amount of lymphatic flow required to transport increases while there is no increase in the muscle pumping function. However, the capability of lymphatic vessels is limited, which leads to cause the relatively overloaded lymphatic fluid.

We receive the impression that the obesity induced lymphedema starts from distal part like age-related lymphedema through our experience at present, however, further investigation is necessary.

Recurrent cellulitis in the lower extremities is common in the elderly [43,44]. Decreased contractile activity in the lymphatics was suspected in the elderly patients with edema in our study. Many factors contribute to edema and cellulitis in this age group; however, it is assumed that lymphatic deterioration contributes to these symptoms, because the condition of the lymphatics is closely related to immune status and a lymphedematous limb is immunocompromised [45]. LVA is a promising surgical treatment to prevent cellulitis in patients with a lymphatic disorder [46,47]. Compression therapy is likely to be burdensome for elderly people [48] and LVA would decrease this burden and the risk of infection in this age group.

## 5. Conclusions

The characteristics of age-related lymphedema are bilateral involvement and deterioration that starts distally rather than proximally. Furthermore, the lymphatics in patients with age-related lymphedema are ectatic, which means that the detection rate is high. LVA is easier to perform. LVA may be more effective in older patients with age-related lymphedema than in their younger counterparts with primary lymphedema.

## Figures and Tables

**Figure 1 jcm-10-05129-f001:**
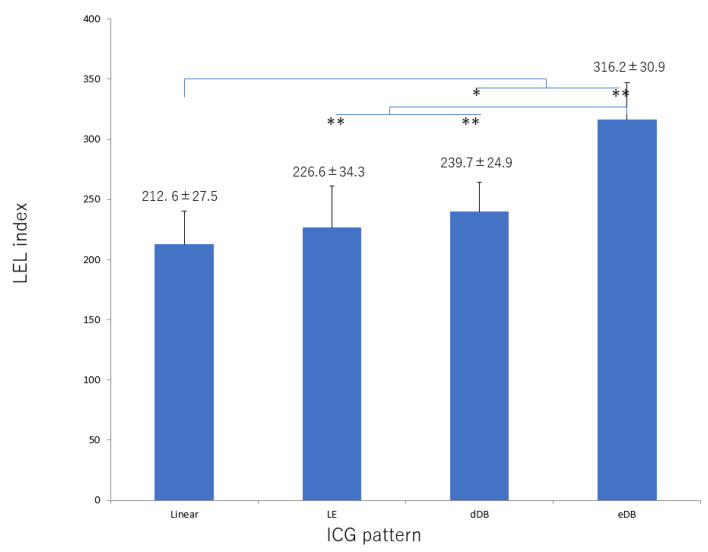
LEL index score according to type of lymphography pattern in the group ≥65 years of age. There was significant deterioration in this score with progression from the linear, LE, dDB through to the eDB pattern. dDB, distal dermal backflow; eDB, extended dermal backflow; LE, low enhancement; LEL, lower extremity lymphedema. * *p* < 0.05, ** *p* < 0.01 (Tukey–Kramer test).

**Figure 2 jcm-10-05129-f002:**
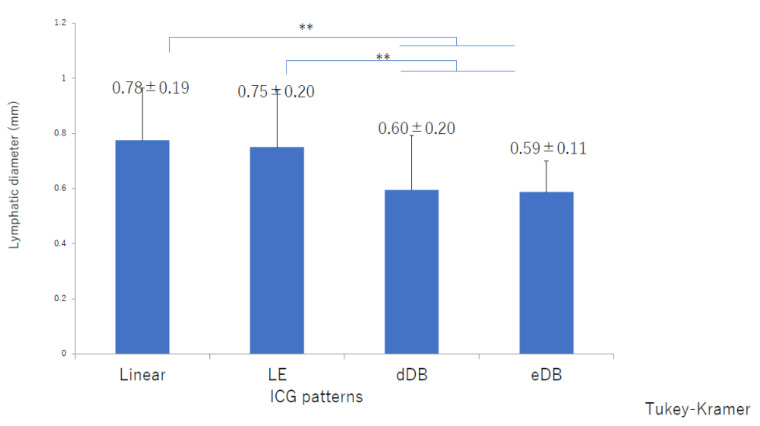
Lymphatic diameters by indocyanine green pattern in the ≥65 group. There was a significant decrease in lymphatic diameter with progression from the linear, LE, dDB through to the eDB pattern. dDB, distal dermal backflow; eDB, extended dermal backflow; LE, low enhancement; LEL, lower extremity lymphedema ** *p* < 0.01 (Tukey–Kramer test).

**Figure 3 jcm-10-05129-f003:**
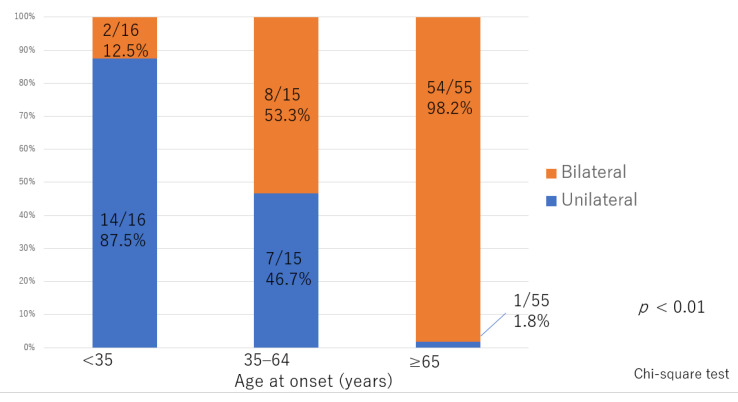
Distribution of laterality according to age of onset. The between group difference was statistically significant.

**Figure 4 jcm-10-05129-f004:**
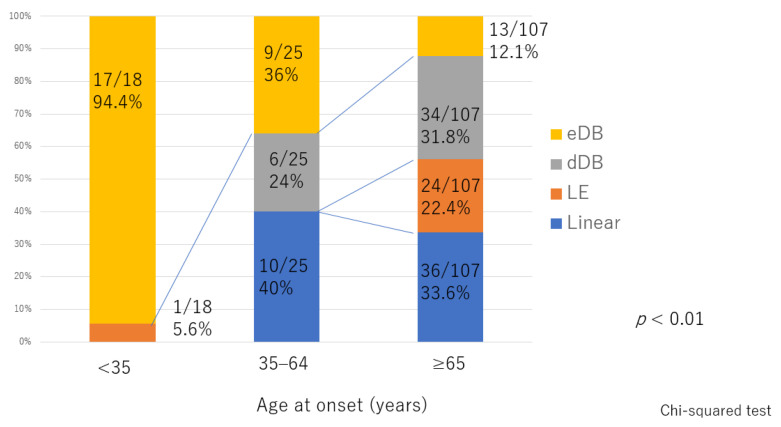
Distribution of indocyanine green pattern according to age of onset. The between-group difference was statistically significant. dDB, distal dermal backflow; eDB, extended dermal backflow; LE, low enhancement.

**Figure 5 jcm-10-05129-f005:**
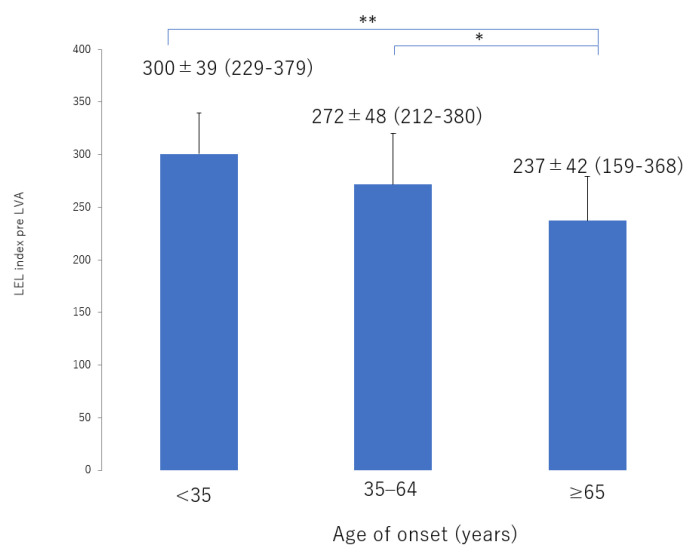
LEL index score before LVA according to age of onset. There was statistically significant deterioration in score with age of onset from the ≥65 group, 35–64 group, through to the <35 group. LEL, lower extremity lymphedema; LVA, lymphaticovenous anastomosis * *p* < 0.05, ** *p* < 0.01 (Tukey–Kramer test).

**Figure 6 jcm-10-05129-f006:**
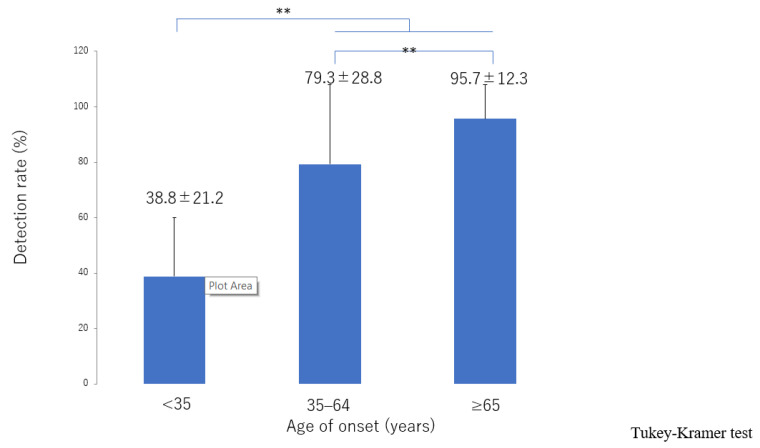
Lymphatic detection rate according to age of onset. The detection rate increased significantly with increasing age of onset from the <35 group, 35–64 group, through to the ≥65 group. ** *p* < 0.01 (Tukey–Kramer test).

**Figure 7 jcm-10-05129-f007:**
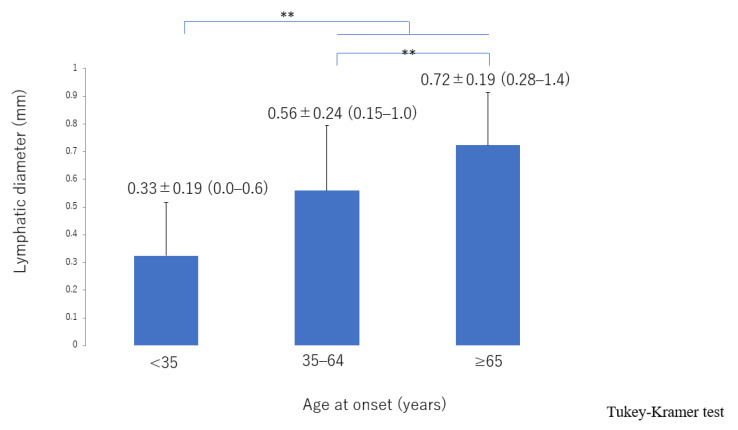
Lymphatic diameter according to age of onset. The lymphatic vessels became significantly more dilated with increasing age of onset from the <35 group, 35–64 group, through to the ≥65 group. ** *p* < 0.01 (Tukey–Kramer test).

**Figure 8 jcm-10-05129-f008:**
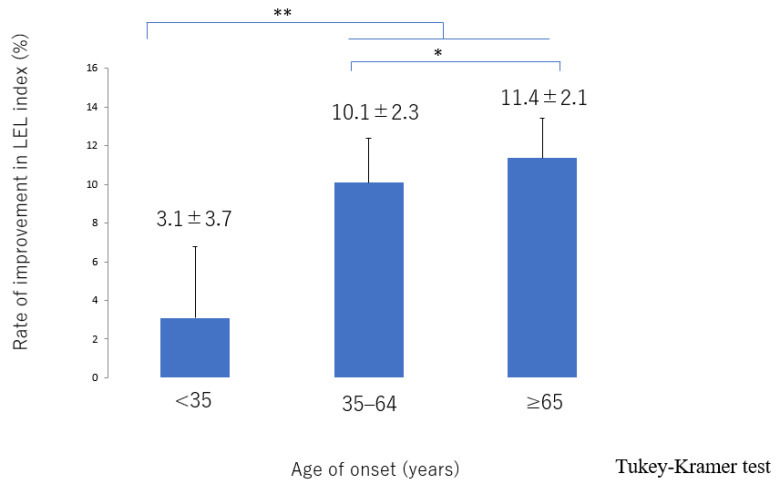
Rate of improvement in the LEL index score according to age of onset. The improvement was significantly greater in the 35–64 and ≥65 groups than in the <35 group and was greater in the ≥65 group than in the 35–64 group. LEL, lower extremity lymphedema * *p* < 0.05, ** *p* < 0.01 (Tukey–Kramer test).

**Figure 9 jcm-10-05129-f009:**
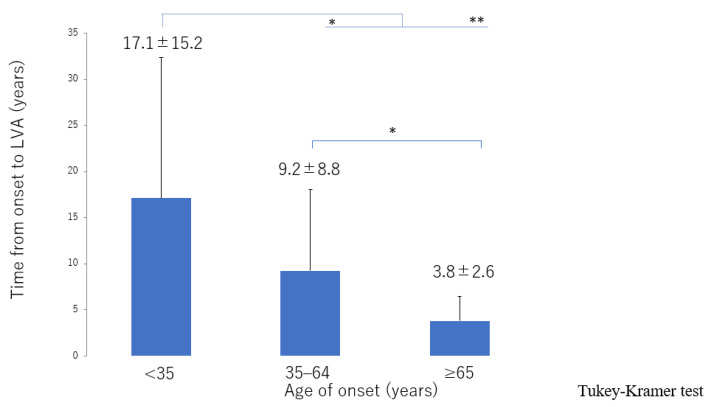
Interval between onset of lymphedema and LVA surgery according to age of onset. The interval decreased from the ≥65 group, 35–64 group, through to the <35 group. LVA, lymphaticovenous anastomosis * *p* < 0.05, ** *p* < 0.01 (Tukey–Kramer test).

**Table 1 jcm-10-05129-t001:** Patient demographics and clinical details of 86 patients with primary lymphedema affecting 150 lower limbs.

Age, years		67.5±19.2 (15–95)			
Sex	Female	56 (65.1%)			
	Male	30 (34.9%)			
Body Mass Index		25.42±4.76 (14.39–34.72)			
Laterality of LEL, *n*	Unilateral	22 (25.6%)			
	Bilateral	64 (74.4%)			
Age at Onset of LEL (years)		60.12 ± 23.27 (0–93)			
Duration of Edema (years)		7.39 ± 9.23 (0.2–53)			
ISL Stage	Stage Ⅰ	*n* = 64 (42.7%)			
	Stage Ⅱ	*n* = 66 (44%)			
	Stage Ⅲ	*n* = 20 (13.3%)			
LEL Index Value		256 ± 49 (159–380)			
ICG Pattern (Limbs)	Linear	LE	dDB		eDB
	46	25	40		39
Number of Patients by Age at Onset	<35 years	35-64 years		≥65 years	
	*n* = 16	*n* = 15		*n* = 55	
Age at LVA (years)	35.6 ± 17.6 (15–72)	59.4 ± 8.5 (44–73)		79.3 ± 5.0 (70–95)	
Observation Period, years		1.6 ± 0.6 (0.8–2.6)			

ISL, International Society of Lymphology; LEL, Lower Extremity Lymphedema; dDB, distal dermal backflow; eDB, extended dermal backflow; ICG, indocyanine green; LE, low enhancement.

## Data Availability

The data presented in this study are available on request from the corresponding author.

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
