# Peer review of "Lymphaticovenous Anastomosis for Age-Related Lymphedema"

_jcm, 2021, doi:10.3390/jcm10215129_

Round 1

Reviewer 1 Report

Interesting paper.

The introduction needs to be better described, especially because the suggestion is made that age-related lymphedema is primary lymphedema. Why don't call it secundary lymphedema caused by age. What about obesity induced lymphedema? What about cellulitis-induced lymphedema?

Normally you don't mention number of patients in the method section but in the result section

How can you take into account the BMI or change of muscle volume if you compare preoperative with postoperative circumference measurements? Because you also included bilateral disease you can't compare both sides.

It is not because age-related lymphedema starts distally that you can conclude that this is a primary lymphedema. indeed in secundary lymphedema casused by destructionof the transport proximal of the leg, the lymphedema starts to develop proximal, but what about obesity-related lymphedema or post-infection?

You mention cellulitis in the discussion section, do you have data of cellulitis in  your patients? 

Author Response

Reviwer1

  1. The introduction needs to be better described, especially because the suggestion is made that age-related lymphedema is primary lymphedema. Why don't call it secundary lymphedema caused by age. What about obesity induced lymphedema? What about cellulitis-induced lymphedema?

We have deleted some words “primary” in Introduction. In general, lymphedema cause of which is clear and is accepted widely is called “secondary”. However, at this moment, we think “age-related”, “obesity induced”, or “cellulitis-induced” is not accepted widely. In future, there may be possibility those lymphedemas are included into secondary lymphedema. But, in this article, we judged it appropriate to include those lymphedemas into primary lymphedema. 

2.Normally you don't mention number of patients in the method section but in the result section

We have corrected it.

3.How can you take into account the BMI or change of muscle volume if you compare preoperative with postoperative circumference measurements? Because you also included bilateral disease you can't compare both sides.

LEL index is the measuring method in which BMI has been already taken into account. We must admit that muscle volume should be taken into account for accurate investigation. But all the patients in the study did not have problems of muscle weakness, the skewness did not affect the results

4.It is not because age-related lymphedema starts distally that you can conclude that this is a primary lymphedema. indeed in secundary lymphedema casused by destructionof the transport proximal of the leg, the lymphedema starts to develop proximal, but what about obesity-related lymphedema or post-infection?

We have added the following sentences into Discussion.

“According to previous reports, the diameter of lymphatic was significantly greater in the obese group than in the nonobese group (40). The results are consistent with studies in hypercholesterolemic mice (41,42). The mechanism how obesity impairs the lymphatic function was unknown. However, a hypothesis is that the amount of lymph produced by obesity patients increases as BMI increases and the amount of lymphatic flow required to transport increases while there is no increase in the muscle pumping function. But the capability of lymphatic vessels is limited, which leads to cause the relatively overloaded lymphatic fluid. At this moment, there has been the impression that obesity induced lymphedema starts from distal part like age-related lymphedema in our experience, how-ever, further investigation is necessary.”

Post-infection is unclear at this moment, further investigation is necessary for us.

5.You mention cellulitis in the discussion section, do you have data of cellulitis in your patients?

We have to investigate from now on. But at this time, it is unclear.

Reviewer 2 Report

In this paper the authors looked at how the age of onset of lower extremity lymphedema was related to the outcomes from LV bypass. The authors found that onset at the oldest age group (65+ years) was most associated with worse progression of lymphedema, greater lymphatic vessel ectasia, and greatest improvement following LV bypass.

The manuscript is clearly written and mostly easy to understand. The figures are of excellent quality. Only minor grammatical errors are seen (for example there should be a period at the end of the Introduction section).

How many patients were initially screened for the study? It would be useful to know how many patients were excluded based on all of the criteria listed. Since this paper is emphasizing the differences in age brackets, it would be good to know how many patients were excluded from each of these age groups.

What is the definition of “major pelvic surgery?” For example, would inguinal lymphadenectomy meet this criteria? In other words is it possible the younger-onset group had a different set of potential causes of lymphedema than the older-onset group?

In many centers, the diagnosis of lymphedema requires demonstration of some leakage- whether it be dermal backflow or non-linear pattern on ICG. The authors chose to include patients with only delayed transit even if the channels are linear. Figure 4 shows that the distribution of patients with lymphedema and linear ICG skews towards the mid and older age groups.

The LEL index is a volume-based measurement. This is known to correlate with other clinical markers for lymphedema, but body volume distribution should differ between different age groups. Can the authors comment on whether this may skew some of the results when comparing young individuals to older ones at the time of measurement?

One of the conclusions that the authors reach is that age-related lymphedema favors development of distal edema first, based on the results summarized in Figure 4. However, the time of evaluation/surgery in the youngest cohort occurred far later than the in the 65+ year group, as summarized in Figure 9. Patients with lymphedema diagnosed at <35 years did not get their measurements/surgeries an average of 17 years after initial occurrence, while those in the oldest cohort underwent evaluation within an average of 3 years.  Shouldn’t we expect there to be more diffuse disease after a longer period of disease, so how can we definitely say that patients who had lymphedema starting at a younger age did not also initially start with distal swelling? There do not appear to be data here to draw that conclusion.

Because measurements were not performed at the time of onset of lymphedema, I am not sure if we can conclude that patients with onset at 35 have smaller lymphatic diameters compared to those with onset at age 65. Again, these measurements are being performed at the time of surgery, not at the time of diagnosis. I would expect that in very chronic cases superficial lymphatic channels damaged over time would involute.  Also, in cases of late onset lymphedema, if we are using a definition of lymphedema which includes lymphatic delay with linear channels and no evidence of backflow, this would also bias the data towards patients with dilated channels.

Author Response

Reviewer2

1.How many patients were initially screened for the study? It would be useful to know how many patients were excluded based on all of the criteria listed. Since this paper is emphasizing the differences in age brackets, it would be good to know how many patients were excluded from each of these age groups.

We have added the following sentences at the beginning of results and supplemental table, “This study included an initial 490 patients with lower extremity lymphedema who underwent lymphedema surgery at our institution between May 2017 and December 2018. Among them, 4 patients in the <35 group, 189 patients in the 35-64 group, and 211 patients in the ≥65 group were excluded by the exclusion criteria (Supplemental table).”.

 2.What is the definition of “major pelvic surgery?” For example, would inguinal lymphadenectomy meet this criteria? In other words, is it possible the younger-onset group had a different set of potential causes of lymphedema than the older-onset group?

The definition is history of a major invasive procedure to treat pelvic cancer including uterine cancer, prostate cancer, rectal cancer, metastatic cancer, and inguinal lymphadenectomy. We have added those description. We think the younger-onset group had a different set of potential causes of lymphedema than the older-onset group

 3.In many centers, the diagnosis of lymphedema requires demonstration of some leakage- whether it be dermal backflow or non-linear pattern on ICG. The authors chose to include patients with only delayed transit even if the channels are linear. Figure 4 shows that the distribution of patients with lymphedema and linear ICG skews towards the mid and older age groups.

We examined any potential causes for leg edema but found no causes except for delayed transit time. Therefore, we judged delayed lymphatic transit time is cause of edema, and decide to include into lymphedema. We have added the following sentence into Patients and Methods, “A diagnosis of lymphedema was made based on an abnormal lymphoscintigram with delayed transit of radiolabeled colloid (i.e., longer than 50 min) (13−17) considering the existence of lower limb edema since there are no causes for edema except for delayed lymphatic transport..”

 4.The LEL index is a volume-based measurement. This is known to correlate with other clinical markers for lymphedema, but body volume distribution should differ between different age groups. Can the authors comment on whether this may skew some of the results when comparing young individuals to older ones at the time of measurement?

LEL index is the measuring method in which BMI has been already taken into account. We must admit that muscle volume should be taken into account for accurate investigation. But all the patients in the study did not have problems of muscle weakness, the skewness did not affect the results

 5.One of the conclusions that the authors reach is that age-related lymphedema favors development of distal edema first, based on the results summarized in Figure 4. However, the time of evaluation/surgery in the youngest cohort occurred far later than the in the 65+ year group, as summarized in Figure 9. Patients with lymphedema diagnosed at <35 years did not get their measurements/surgeries an average of 17 years after initial occurrence, while those in the oldest cohort underwent evaluation within an average of 3 years.  Shouldn’t we expect there to be more diffuse disease after a longer period of disease, so how can we definitely say that patients who had lymphedema starting at a younger age did not also initially start with distal swelling? There do not appear to be data here to draw that conclusion.

We are sorry, there may be some misunderstanding. We are not saying that patients who had lymphedema starting at a younger age did not initially start with distal swelling. We are just saying that patients who had lymphedema starting at a younger age have features of unilateral and severity and those features correspond with secondary lymphedema. We think we did not mention about the beginning features of younger patients.

6. Because measurements were not performed at the time of onset of lymphedema, I am not sure if we can conclude that patients with onset at 35 have smaller lymphatic diameters compared to those with onset at age 65. Again, these measurements are being performed at the time of surgery, not at the time of diagnosis. I would expect that in very chronic cases superficial lymphatic channels damaged over time would involute.  Also, in cases of late onset lymphedema, if we are using a definition of lymphedema which includes lymphatic delay with linear channels and no evidence of backflow, this would also bias the data towards patients with dilated channels.

The reviewer’s suggestion is reasonable and may be right. The authors also mentioned the possibility same as the reviewer. However, there may be some shortage of emphasis. We have corrected the following sentences in discussion.

“Second, the lymphatic diameter was greater in older patients in the biggening stage than in younger patients or older patients in the advanced stage.”, “In the present study, patients <35 years of age had the most severe disease features and the least favorable surgical outcomes. Furthermore, our younger patients had longer interval between disease onset and LVA. This pronounced tendency has yet to be explained but may lead to smaller diameter of lymphatics and worse outcomes owing to long lasting chronic damage to lymphatics.”

Round 2

Reviewer 1 Report

No further comments

Author Response

Thank you very much. We are happy!

Reviewer 2 Report

The authors have made clarified most of the points that were raised during the first review and are satisfactory.

Following this revision there are many new spelling/grammatical errors that require fixing. For example on page 11 line 242 "biggening" should be spelled "beginning" but a more appropriate term should be "early".  

Author Response

Thank you very much. We have made some corrections.